# Microstructures and Properties of the Low-Density Al$_{15}$Zr$_{40}$Ti$_{28}$Nb$_{12}$M(Cr, Mo, Si)$_5$ High-Entropy Alloys

Yasong Li [1], Peter K. Liaw [2] and Yong Zhang [1,3,4,*]

1 Beijing Advanced Innovation Center of Materials Genome Engineering, State Key Laboratory for Advanced Metals and Materials, University of Science and Technology Beijing, Beijing 100083, China; songli159@sina.com

2 Department of Materials Science and Engineering, The University of Tennessee, Knoxville, TN 37996, USA; pliaw@utk.edu

3 Qinghai Provincial Key Laboratory of New Light Alloys, Qinghai Provincial Engineering Research Center of High Performance Light Metal Alloys and Forming, Qinghai University, Xining 810016, China

4 Shunde Graduate School, University of Science and Technology Beijing, Foshan 528399, China

* Correspondence: drzhangy@ustb.edu.cn

**Abstract:** Low-density materials show promising prospects for industrial application in engineering, and have remained a research hotspot. The ingots of Al$_{15}$Zr$_{40}$Ti$_{28}$Nb$_{12}$Cr$_5$, Al$_{15}$Zr$_{40}$Ti$_{28}$Nb$_{12}$Mo$_5$ and Al$_{15}$Zr$_{40}$Ti$_{28}$Nb$_{12}$Si$_5$ high-entropy alloys were prepared using an arc melting method. With the addition of the Cr, Mo, and Si, the phase structures of these alloys changed to a dual phase. The Cr and Mo promote the formation of the B2 phase, while the Si promotes the formation of a large amount of the silicides. The compression yield strengths of these alloys are ~1.36 GPa, ~1.27 GPa, and ~1.35 GPa, respectively. The addition of Si and Cr significantly reduces the compression ductility, and the Al$_{15}$Zr$_{40}$Ti$_{28}$Nb$_{12}$SiMo$_5$ high-entropy alloy exhibits excellent comprehensive mechanical properties. This work investigated the influence of Cr, Mo, and Si on the phase structures and properties of the low-density Al-Zr-Ti-Nb high-entropy alloys, providing theoretical and scientific support for the development of advanced low-density alloys.

**Keywords:** low-density; high-entropy alloys; microstructures; properties; phase structures

## 1. Introduction

Low-density design plays a crucial role in the development and progression of the next-generation structural materials with high-performance, efficiency gains and environmental friendliness. High-entropy alloys (HEAs) or multi-component alloys (MCPAs) have revolutionized the design strategy of the traditional alloy, and attracted considerable attention due to their attractive comprehensive mechanical properties [1–6]. Nowadays, a wide variety of HEAs are available, including face-centered-cubic (FCC) HEAs with excellent ductility [7–12], body-centered-cubic (BCC) HEAs with high strength [13–18], and hexagonal-close-packed structure (HCP) HEAs combined with rare-earth alloy elements [19–21], or transformation-induced-plasticity (TRIP) [22]. These investigations demonstrate the great research prospects of HEAs. However, more work still needs to be done to achieve advanced HEAs with higher strengths.

BCC HEAs display a relatively high intrinsic yield strength with the addition of refractory elements, such as NbMoTaW [13,16], and TiZrHfNbTa [23,24], etc. The inherent high density of these elements limited the development of these alloy systems. Therefore, several studies have been focusing on the development of low-density HEAs. The AlLiMg-based HEAs were reported to have multi-phase structures due to the negative effect on the binary mixing enthalpy ($\Delta H_{mix}$) [25–30]. Only the AlLiMgScTi low-density HEA formed an FCC solid solution structure via mechanical alloying [26]. Furthermore, another type of the low-density HEAs is implemented for lightweight refractory HEAs with BCC structures.

These alloys mainly contain refractory elements, such as the Ti, Zr, Nb, and V, etc., and Al, Cr, and Si elements, etc. Yang et al. [31] investigated the effect of Al element on the NbTiVTaAl$_x$ HEAs alloys, which present BCC structures with excellent compression ductility exceeding 50% without the breakage. Among them, the NbTiVTaAl$_{0.25}$ HEA shows the highest yield strength at around 1330 MPa. In addition, a low-density Cr-Nb-Ti-V-Zr system was also investigated. With the addition of low-density refractory alloy elements, the NbTiVZr and NbTiV$_2$Zr HEAs form disordered BCC structures, and the Cr element promotes the formation of the ordered Laves phase, thereby improving the hardness of these alloys [32,33]. The addition of the Al element can enhance the yield strength of the Al$_x$NbTiMoV HEAs [34], and Al is an element that can increase the stability of BCC or B2 structures [35,36]. In addition, the AlTiNbV, AlNbTiVZr$_{0.5}$, and AlTiVCr low-density HEAs display single BCC structures, high specific strengths, and an improved ductility with the addition of Zr [36–38].

The Cr and Mo can improve the high-temperature properties of the BCC HEAs [39,40], and the Si can also promote some Laves phase to enhance the properties [41]. Several studies have indicated that the addition of Al can increase the strength of the Zr$_{50}$Ti$_{35}$Nb$_{15}$ alloy, and the Al$_x$(Zr$_{50}$Ti$_{35}$Nb$_{15}$)$_{100-x}$ HEAs present high strength and excellent plasticity [15,42]. Therefore, the present work investigates the effect of the Cr, Mo, and Si on microstructures and properties of the Al-Zr-Ti-Nb HEAs, and attempts to demonstrate the influence of these elements on the phase formation of low-density BCC HEAs.

## 2. Materials and Methods

The ingots of Al$_{15}$Zr$_{40}$Ti$_{28}$Nb$_{12}$Cr$_5$ (Cr$_5$), Al$_{15}$Zr$_{40}$Ti$_{28}$Nb$_{12}$Mo$_5$ (Mo$_5$), and Al$_{15}$Zr$_{40}$Ti$_{28}$Nb$_{12}$Si$_5$ (Si$_5$) HEAs were prepared by the vacuum arc smelting under an Ar atmosphere, and each sample of approximately 40 g was melted at least five times with pure Al, Zr, Ti, Nb, Cr, Mo and Si (99.95 wt.%). The phase structure of the samples was determined by X-ray Diffraction (XRD) using Rigaku DMAX-RB Cu K$\alpha$ radiation with scattering angles in the range of 10–90° and a scanning rate of 5°/min. The microstructure and fracture cross-sectional morphology were characterized by a Zeiss Supra 55 filed emission scanning electron microscopes (SEM, SUPRA 55, Carl Zeiss AG, Jena, Germany), equipped with an energy-dispersive X-ray spectrometer (EDS, UltraDry EDS, Thermo Scientific™, Waltham, MA, USA) and electron backscattered diffraction (EBSD, C-swift, Oxford Instruments plc, Tubney Woods, Abingdon, UK,). The samples for SEM observation were treated by mechanical polishing, and those for the EBSD test were mechanically ground by a 3500-grit SiC paper, followed by electro-chemically polishing with a mixture of 80% ethanol and 20% perchloric acid (vol.%) at room temperature [43]. The density of the alloy was measured based on the Archimedes Principle, with the following measurement formula:

$$\rho_{measured} = \frac{m_1}{m_2} \cdot \rho_{H_2O} \tag{1}$$

where $\rho_{measured}$ is the measured density of the alloy, $m_1$ is the mass of the sample in air, $m_2$ is the mass gain of water when the sample is immersed in distilled water, and $\rho_{H_2O}$ is the density of water.

The samples for the compression test were cut from the ingot with a size of Φ3 × 6 mm. The room-temperature compression test was conducted by a CMT4105 universal electronic testing machine (Suns, Shenzhen, China) with an initial strain rate of $2.0 \times 10^{-4}$ s$^{-1}$. Three samples were tested with the same composition.

## 3. Results

### 3.1. The Microstructures and the Phase Sructures

Figure 1 shows the SEM images with the back-scattered electron (BSE) of Cr$_5$, Mo$_5$, and Si$_5$ HEAs. It can be seen from Figure 1a,b that with the addition of Cr and Mo, the Cr$_5$ and Mo$_5$ HEAs display a single solid solution phase. The BSE microstructure shown in Figure 1c indicates that there is a large number of dark phases (DPs) appearing in the solid

solution matrix of the Si$_5$ HEA. These DPs present a needle-like sharp eutectic HEAs [44] structure with BCC + DP and a mesh-shape distribution in the matrix, suggesting that the addition of Si promotes the formation of an ordered silicide phase [41,45]. This structure guarantees that this alloy has an excellent casting property.

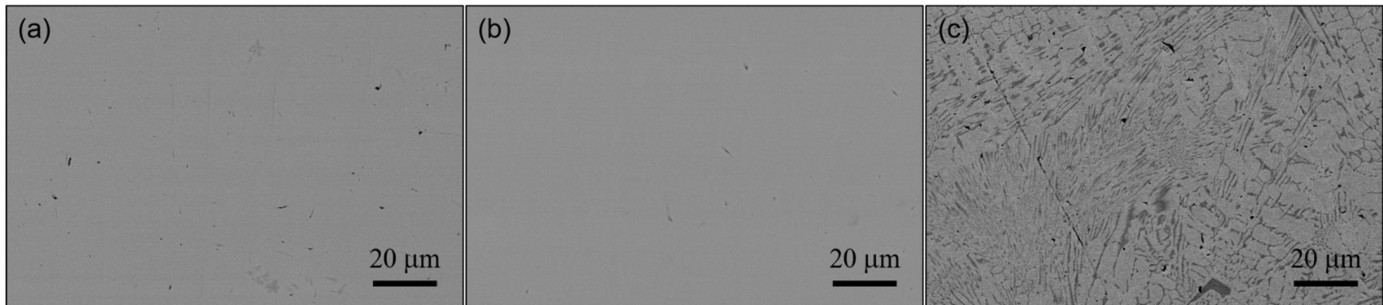

**Figure 1.** The SEM images of the Al$_{15}$Zr$_{40}$Ti$_{28}$Nb$_{12}$M$_5$(Cr, Mo, Si) HEAs with BSE. (**a**) Cr$_5$ HEAs, (**b**) Mo$_5$ HEAs, and (**c**) Si$_5$ HEAs.

The EDS mapping of the Al$_{15}$Zr$_{40}$Ti$_{28}$Nb$_{12}$M$_5$(Cr, Mo, Si) HEAs is displayed in Figure 2. As can been seen from Figure 2a,b, the distribution of each element is still uniform in the Cr$_5$ and Mo$_5$ HEAs alloys. However, with the addition of Si, a large number of Si-rich precipitates appeared in the matrix. In addition, these precipitates are poor in Ti, as shown in Figure 2c, and the Al, Zr, and Nb display uniform distributions.

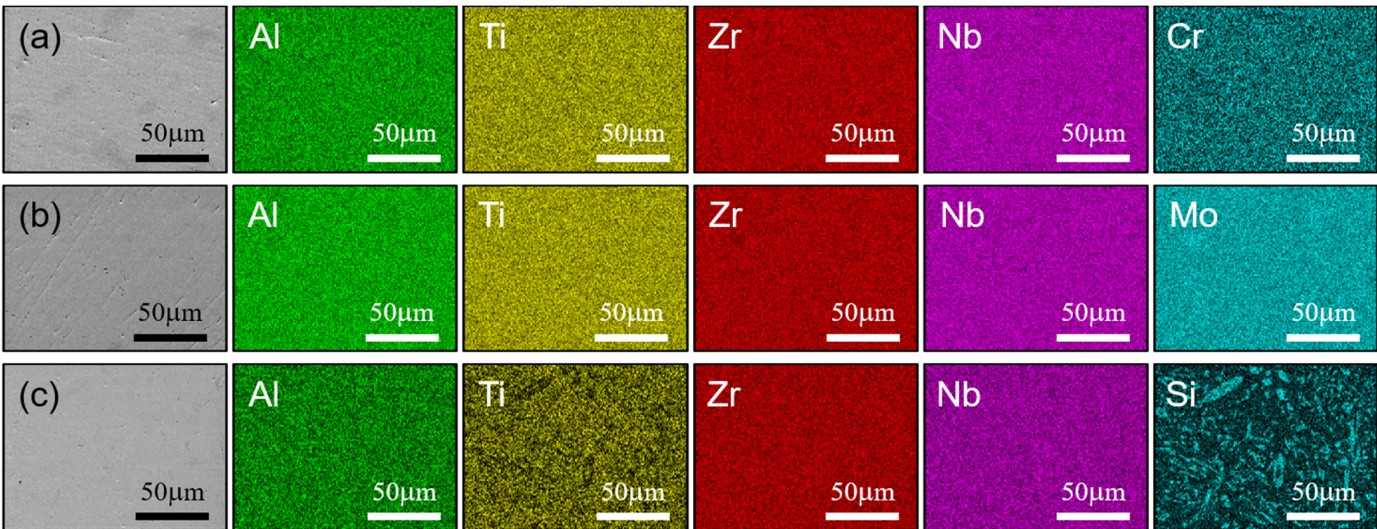

**Figure 2.** The EDS mappings of the Al$_{15}$Zr$_{40}$Ti$_{28}$Nb$_{12}$M$_5$(Cr, Mo, Si) HEAs with secondary electron (SE2). (**a**) Cr$_5$ HEA, (**b**) Mo$_5$ HEA, and (**c**) Si$_5$ HEA.

The XRD patterns of the Al$_{15}$Zr$_{40}$Ti$_{28}$Nb$_{12}$M$_5$(Cr, Mo, Si) HEAs are shown in Figure 3. It can be seen that the main phase of these alloys is a BCC solid solution structure. A small amount of B2 structure appears with the addition of Cr and Mo. A large amount of silicide appears in the Si$_5$ HEA with the addition of Si, which exhibits DP in Figure 1c. The peak at 26° in the XRD patterns of these HEAs indicates that some ordered B2 phases have been formed in these alloys [44,46], and there is also a peak with illustration enlargement with the pattern of the Mo$_5$ alloy. According to the reported Al10 (Al$_{10}$(Zr$_{50}$Ti$_{35}$Nb$_{15}$)$_{90}$) [15] and Zr$_{50}$Ti$_{35}$Nb$_{15}$ [42] alloy, the ordered B2 phase indicates that the Cr, Mo, Si, and Al can promote the formation of an ordered BCC structure in the Zr-Ti-Nb-based alloys.

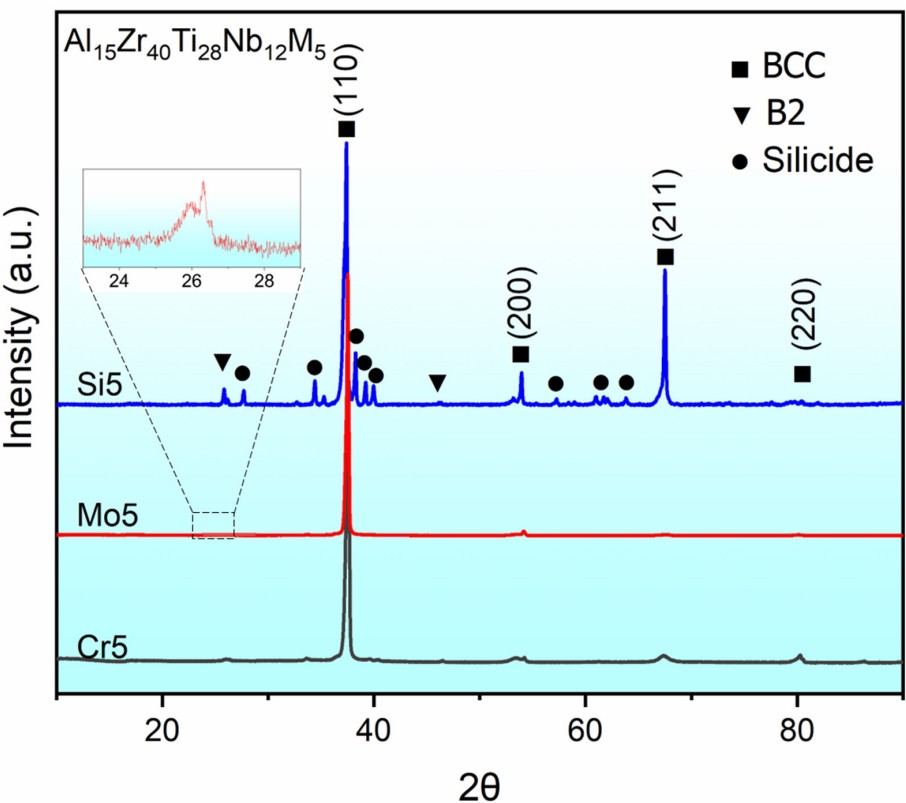

**Figure 3.** The XRD pattern of the $Al_{15}Zr_{40}Ti_{28}Nb_{12}M_5$(Cr, Mo, Si) HEAs.

The EBSD images of the $Cr_5$ and $Mo_5$ HEAs with all Euler angles are shown in Figure 4a,b, respectively. The BCC structure with a lattice constant of Zr was applied to determine the lattice of these HEAs with a step of 2.5 μm. However, the lattice of Zr alloy can also be used for the resolution of these B2 phases. Additionally, the B2 and BCC phase cannot be discriminated between using EBSD. We find that these two alloys generate equiaxed grains, and the grain size of the $Cr_5$ and $Mo_5$ HEAs with the EBSD data are presented in Figure 4c,d, respectively. The grain size of the $Cr_5$ HEA ranges from 15 to 125 μm, with a concentrated distribution of 15–40 μm and average grain size of ~28.0 μm. In addition, the grain size of the $Mo_5$ HEA ranges from 15 to 95 μm, with a concentrated distribution of 15–55 μm and average grain size of ~33.8 μm. Due to a large amount of silicides in the microstructure of $Si_5$ HEA, it is hard to obtain grain sizes or EBSD images of the $Si_5$ alloy. Since the smelting method of these alloys is similar, and a large number of precipitations occur during solidification, we speculate that the grain size of the $Si_5$ alloy is smaller than the $Cr_5$ and $Mo_5$ alloys.

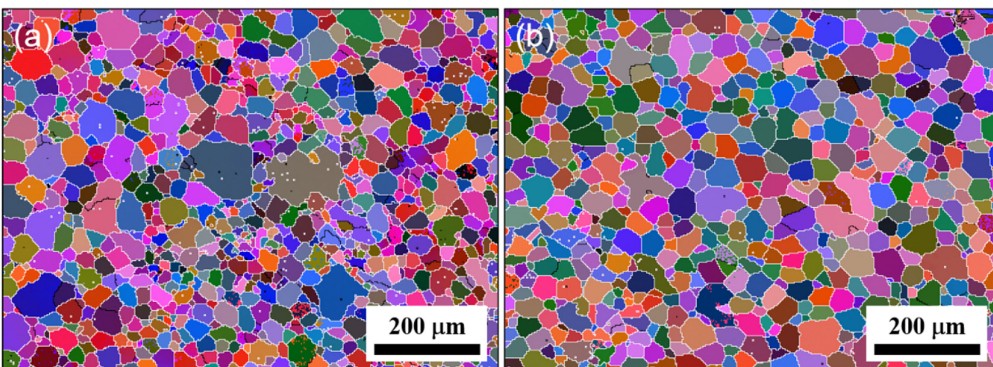

**Figure 4.** *Cont.*

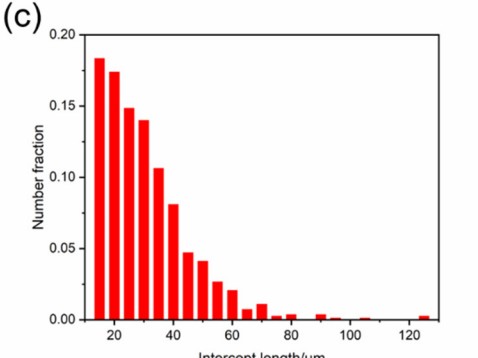 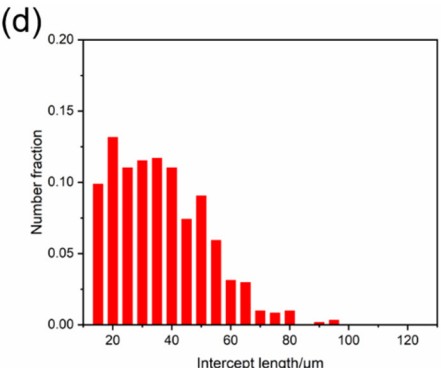

**Figure 4.** The EBSD images with all Euler angles and the grain size obtained by the EBSD technique of these HEAs, (**a,b**) the EBSD images with all Euler angles of $Cr_5$ and $Mo_5$ HEAs, (**c,d**) the grain size of the $Cr_5$ and the $Mo_5$ HEAs.

### 3.2. Properities and Compression Fracture Mechanism

The compressive engineer stress-engineering strain curves of the $Al_{15}Zr_{40}Ti_{28}Nb_{12}M_5$ (Cr, Mo, Si) HEAs are displayed in Figure 5. The yield strengths of the $Cr_5$, $Mo_5$, and $Si_5$ HEAs are ~1.36 GPa, ~1.27 GPa, and ~1.35 GPa, respectively, as displayed in Figure 5. Moreover, the compressive plasticity values of these alloys are ~9%, over 50%, and ~7%, respectively. Compared with the Al10 alloys reported in Ref. [15], the compressive yield strengths of the $Al_{15}Zr_{40}Ti_{28}Nb_{12}M_5$(Cr, Mo, Si) HEAs are distinctly enhanced from ~1 GPa to ~1.3 GPa. In addition, the yield strength of these alloys are similar to that of the Al20 ($Al_{20}(Zr_{50}Ti_{35}Nb_{15})_{80}$) alloy [15], suggesting that the strength-hardening effect of the addition of Cr, Mo, and Si is comparable to that of the addition of Al. The properties of the $Mo_5$ alloy are similar to those of the Al20 alloy. However, the ductility drops severely, mainly because the formation of a large number of silicides enhances the strength of the $Si_5$ HEA. The actual densities of these alloys are below 6 g·cm$^{-3}$, with ~5.82, ~6.00, and ~5.79 g·cm$^{-3}$ for $Cr_5$ HEA, $Mo_5$ HEA and $Si_5$ HEA, respectively. The theoretical densities, the actual densities and properties of these alloys are listed in Table 1.

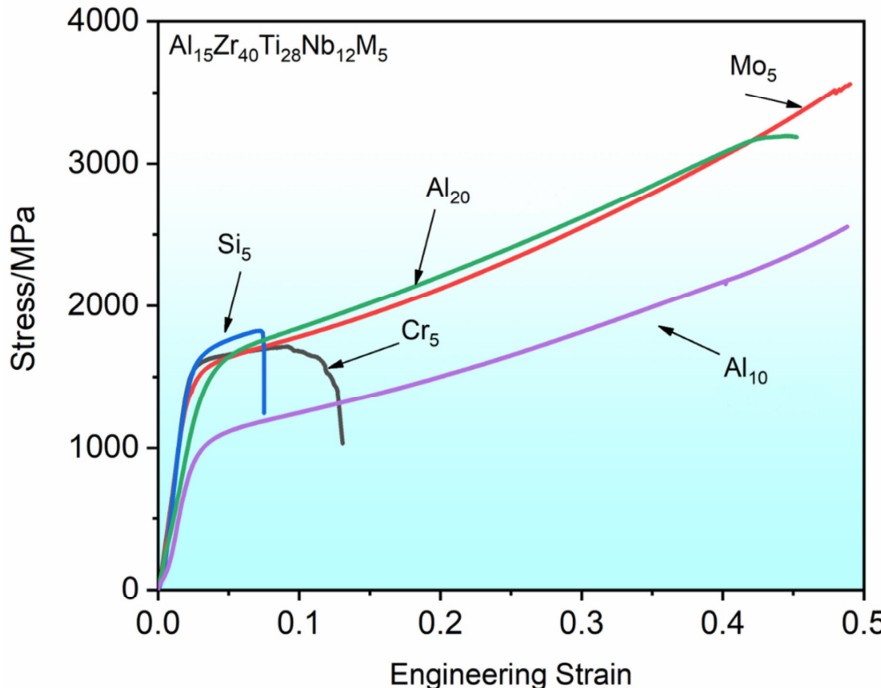

**Figure 5.** Compressed stress–strain curves of the $Al_{15}Zr_{40}Ti_{28}Nb_{12}M_5$(Cr, Mo, Si) HEAs and $Al_x(Zr_{50}Ti_{35}Nb_{15})_{100-x}$ [15] HEAs (reproduced from Ref. [15], with permission from Elsevier, 2022).

**Table 1.** The theoretical densities, actual density and properties of the $Al_{15}Zr_{40}Ti_{28}Nb_{12}M_5$(Cr, Mo, Si) alloys.

| Alloys | Theoretical Density [1] (g·cm⁻³) | Actual Density (g·m⁻³) | Yield Strength (MPa) | Ductility (%) |
|---|---|---|---|---|
| $Al_{15}Zr_{40}Ti_{28}Nb_{12}Cr_5$ | 5.78 | 5.82 | 1357 | 9 |
| $Al_{15}Zr_{40}Ti_{28}Nb_{12}Mo_5$ | 5.92 | 6.00 | 1275 | 50 |
| $Al_{15}Zr_{40}Ti_{28}Nb_{12}Si_5$ | 5.57 | 5.79 | 1346 | 7 |

[1] The theoretical density is calculated by the Formula: $\rho_{theoretical} = \frac{\sum c_i A_i}{\sum c_i A_i / \rho_i}$. Here, $c_i$, $A_i$, and $\rho_i$ are the concentration, atomic weight, and density of the $i$th element, respectively.

The SEM images of the samples after the compression test and the fracture surface of the $Al_{15}Zr_{40}Ti_{28}Nb_{12}M_5$(Cr, Mo, Si) HEAs are shown in Figure 6. Figure 6a–c show the compressed sample morphologies of the $Cr_5$, $Mo_5$ and $Si_5$ HEAs, respectively. It can be found that the alloys exhibit obvious brittle fracture with the addition of Cr and Si. However, the sample of the $Mo_5$ HEA is not completely damaged, with only cracks and slip lines appearing on the sample surface. The fracture-surface images of the $Cr_5$ HEAs are shown in Figure 6d. There are many cleavages planes and river patterns on the fracture surface, indicating that the alloy exhibits significant cleavage fracture. The fracture-surface image of the $Mo_5$ alloy is presented in Figure 6b,e. It can be seen that there are many slip lines and dimples on the surface of the sample and fracture surface, leading to ductile fracture of the $Mo_5$ alloy. The fracture-surface image of $Si_5$ alloy is exhibited in Figure 6f. The river pattern and needle-sheet structure in the fracture suggest that brittle breakage occurs during the progress of compression.

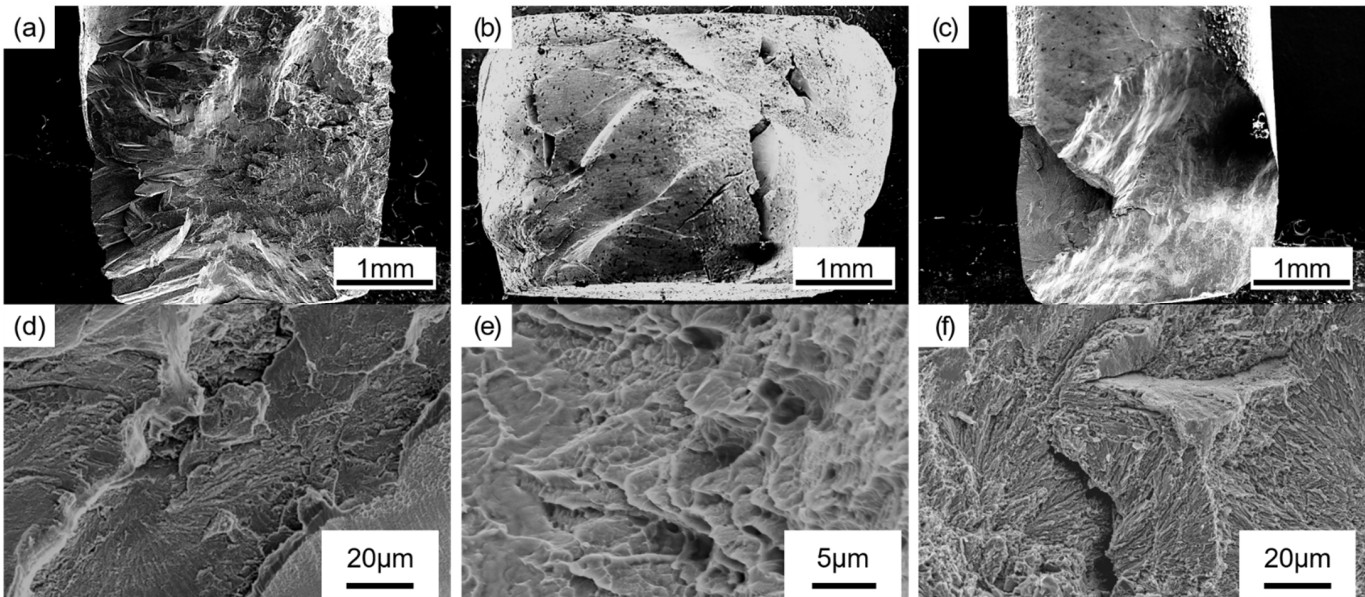

**Figure 6.** The SEM images of the samples after the compression test and the fracture surface of the $Al_{15}Zr_{40}Ti_{28}Nb_{12}M_5$(Cr, Mo, Si) HEAs, (**a**–**c**) the compressed samples morphology of the $Cr_5$, $Mo_5$ and $Si_5$ HEAs, respectively; (**d**–**f**) the fracture-surface image of the $Cr_5$, $Mo_5$ and $Si_5$ HEAs, respectively.

## 4. Discussion

Owing to the component complexity of the HEAs, a higher mixing entropy ($\Delta S_{mix}$) would promote the formation of solid-solution structures in HEAs [34]. Furthermore, these parameters $\Delta H_{mix}$, $\delta$, $\Omega$, and $VEC$, etc. [47–49] provide a clearer and easier method by which to predict the phase formation for the design of HEAs. Here, these parameters are defined.

$$\Delta S_{mix} = -R \sum_{i=1}^{n} c_i \ln c_i,$$
(2)

$$\delta = \sqrt{\sum_{i=1}^{n} c_i \left(1 - \frac{r_i}{\sum_{i=1}^{n} c_i r_i}\right)^2}, \quad (3)$$

$$\Delta H_{mix} = \sum_{i=1,i\neq j}^{n} 4 c_i c_j \Delta H_{mix}^{ij}, \quad (4)$$

$$\Omega = T_m \Delta S_{mix} / |\Delta H_{mix}|, \quad (5)$$

$$VEC = \sum_{i=1}^{n} c_i (VEC)_i, \quad (6)$$

where $R$ is the gas constant, $c_i$ and $c_j$ are the atomic fraction of the $i$th and $j$th elements, $\delta$ is the atomic size difference, $\Delta H_{mix}^{ij}$ denotes the binary mixing enthalpy of the $i$th and $j$th elements [50], $r_i$ is the radius of $i$th element, $\Omega$ is a multi-component solid solution rule, $T_m$ is the average melting point, and $VEC$ is the valence electron concentration. As previously reported, it is easier to form the solid solution with a larger enthalpy and a smaller $\delta$ value [47] under $\Omega \geq 1.1$ and $\delta \leq 6.6\%$. In addition, with the help of the $VEC$, the phase stability for BCC or FCC phases in HEAs can be quantitatively predicted. When the $VEC < 6.87$, the BCC phase is stable in the alloy. When the $VEC \geq 8.0$, the FCC phase is stable. When $6.87 \leq VEC \leq 8.0$, the BCC or FCC phase would coexist.

In the present work, the parameters of $Al_{15}Zr_{40}Ti_{28}Nb_{12}M_5$(Cr, Mo, Si) and $Al_x$ $(Zr_{50}Ti_{35}Nb_{15})_{100-x}$ HEAs [15] are calculated according to Equations (2) and (6). The binary mixing enthalpy of all of the elements are listed in Table 2. We found that the binary mixing enthalpies of the Al, Cr, Mo and Si with other elements are negative, which means that these alloy elements more easily form the ordered phase with other elements. This feature is responsible for the ordered B2 phase formed in these alloys, as indicated by the XRD result in Figure 2. Furthermore, the binary mixing enthalpy of Si is lower than that of Al, Mo and Cr, leading to a large number of silicides appearing in the $Si_5$ alloy. Table 3 lists the corresponding results with the parameters of these alloys. Chen et al. found that the $\Omega$ values of the $Al_x$NbTiMoV HEAs were reduced with the addition of Al [34]. This trend also occurs in $Al_x(Zr_{50}Ti_{35}Nb_{15})_{100-x}$ alloys [15]. With the addition of the Cr and Mo, the $\Omega$ values of these alloys increased compared to that of the Al20 alloy, indicating that the additions of the Cr and Mo elements stabilized the solid-solution phase in the $Al_{15}Zr_{40}Ti_{28}Nb_{12}M_5$ alloy. Additionally, with the addition of Si, the $\Omega$ value decreased to less than 1.1, and the $\delta$ was found to be the largest (7.29 %) in these alloys, which promotes the formation of a multi-phase structure in the $Si_5$ HEA. According to the calculated $VEC$ value of these alloys, we found that the $VEC$ values of these alloys are less than 6.87, suggesting that these alloys could form the BCC structure.

**Table 2.** The binary mixing enthalpy of the elements added in these HEAs [50].

| $\Delta H_{mix}^{ij}$ | Al | Zr | Ti | Nb | Cr | Mo | Si |
|---|---|---|---|---|---|---|---|
| Al | - | −44 | −30 | −18 | −10 | −5 | −19 |
| Zr | - | - | 0 | 4 | −12 | −6 | −84 |
| Ti | - | - | - | 2 | −7 | −4 | −56 |
| Nb | - | - | - | - | −7 | −6 | −37 |

**Table 3.** The calculated parameters for the phase formation of the $Al_{15}Zr_{40}Ti_{28}Nb_{12}M_5$(Cr, Mo, Si) and $Al_x(Zr_{50}Ti_{35}Nb_{15})_{100-x}$ [15] HEAs.

| Alloys | $\Delta S_{mix}$(J·K$^{-1}$·mol$^{-1}$) | $\Delta H_{mix}$(kJ·mol$^{-1}$) | $T_m$(K) | $\delta$ (%) | $VEC$ | $\Omega$ |
|---|---|---|---|---|---|---|
| $Al_{10}(Zr_{50}Ti_{35}Nb_{15})_{90}$ [15] | 10.17 | −11.3598 | 2033.57 | 5.07 | 4.04 | 1.82 |
| $Al_{20}(Zr_{50}Ti_{35}Nb_{15})_{80}$ [15] | 10.80 | −21.3952 | 1911.28 | 5.13 | 3.92 | 0.96 |
| $Al_{15}Zr_{40}Ti_{28}Nb_{12}Cr_5$ | 11.74 | −17.6792 | 1973.63 | 6.30 | 4.07 | 1.31 |
| $Al_{15}Zr_{40}Ti_{28}Nb_{12}Mo_5$ | 11.74 | −16.8572 | 2009.43 | 5.46 | 4.07 | 1.40 |
| $Al_{15}Zr_{40}Ti_{28}Nb_{12}Si_5$ | 11.74 | −27.1732 | 1948.98 | 7.29 | 3.97 | 0.84 |

Excellent mechanical properties were reported by Yan et al. [15], as they found that some B2 particles appeared in the Al10 HEA tensile test samples with the cold rolling and annealing processes. Moreover, the CrNbTiZr and the CrNbTiVZr HEAs present a higher level of hardness than the CrNbTiVZr and NbTiV$_2$Zr HEAs alloys without the addition of Cr [32]. In addition, some studies have found that some Cr-rich Laves phases are precipitated during heat treatment, which enhances the properties of these alloys [33,51–54]. The Mo is a refractory element commonly used to improve the high-temperature properties [13,39,40,55–58], and the Si is a common non-metallic low-density additive element that can form silicides with multiple elements [41,59]. For the Al$_{15}$Zr$_{40}$Ti$_{28}$Nb$_{12}$M$_5$(Cr, Mo, Si) alloys, the yield strength is similar to the Al20 [15] alloy. However, the ductility of the Cr$_5$ and Si$_5$ alloys decrease significantly. In this work, we found that with the addition of the Cr and Si, the $\delta$ values of the Cr$_5$ and Si$_5$ alloys were much larger than the Al$_x$(Zr$_{50}$Ti$_{35}$Nb$_{15}$)$_{100-x}$ alloys. Nevertheless, with the addition of the Mo, the $\delta$ value of the Mo$_5$ alloy is similar to the Al$_x$(Zr$_{50}$Ti$_{35}$Nb$_{15}$)$_{100-x}$ alloys. In addition, the $\Omega$ values of the Cr$_5$ and Mo$_5$ alloys are between those of the Al10 and Al20 alloys, and the $\Omega$ value of the Si$_5$ alloy is the lowest. Therefore, the $\delta$ value plays a greater effect on the ductility of the alloys than the $\Omega$ vale, indicating that a larger $\delta$ value is responsible for the large lattice distortion in facilitating the orderly formation. Besides, this promotes the ordered phase formation, and reduce the ductility. In addition, a 10 kg ingot of the Al20 alloy was prepared by the vacuum magnetic suspension technology, as shown in Figure 7a. The length unit of the scale in the figure is cm. Therefore, the phase structure of this ingot is the same as that of the low mass one [15], as presented in Figure 7b. Subsequently, the 10 kg level ingots of Al$_{15}$Zr$_{40}$Ti$_{28}$Nb$_{12}$M$_5$(Cr, Mo, Si) HEAs can also be prepared by this method, which provides support for research and industrialization application.

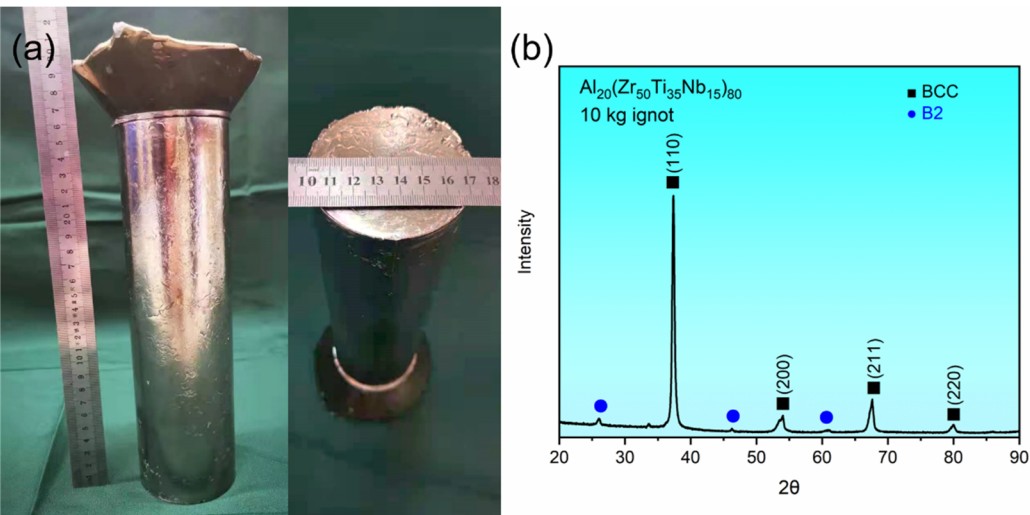

**Figure 7.** (**a**) The 10 kg ingot of Al$_{20}$(Zr$_{50}$Ti$_{35}$Nb$_{15}$)$_{80}$ alloy, (**b**) the XRD pattern of the ingot.

## 5. Conclusions

In this work, the microstructures and properties of the low-density Al$_{15}$Zr$_{40}$Ti$_{28}$Nb$_{12}$M$_5$ (Cr, Mo, Si) HEAs were investigated. The density of these alloy is lower than 6 g·cm$^{-3}$, and the main phase of these alloys belongs to the BCC structure. With the addition of Cr and Mo, some of the B2 phase forms in these alloys, and the Si addition promotes the formation of the silicides. The yield strengths of these alloys are similar, ~1.3 GPa, and the Cr and Si elements exert a negative effect on the ductility due to the large $\delta$ value. However, the addition of Mo has little influence on the properties of the alloy. Furthermore, a 10 kg ingot of the Al$_{20}$(Zr$_{50}$Ti$_{35}$Nb$_{15}$)$_{80}$ alloy was prepared with the same phase structure as that of the low mass one. We anticipate the development of a low-density BCC HEAs to achieve high-temperature applications.

**Author Contributions:** Conceptualization, methodology, validation, formal analysis, investigation, data curation, writing—original draft preparation, Y.L.; resources, writing—review and editing, P.K.L.; resources, writing—review and editing, supervision, project administration, funding acquisition, Y.Z. All authors have read and agreed to the published version of the manuscript.

**Funding:** The present research was funded by the Guangdong Basic and Applied Basic Research Foundation (No.2019B1515120020), the State Key Laboratory for Advanced Metals and Materials in the University of Science and Technology Beijing (No. 2020Z-08), and the Funds for Creative Research Groups of China (No. 51921001); P.K.L greatly thanks the supports from (1) the National Science Foundation (DMR-1611180 and 1809640) and (2) the Army Research Office (W911NF-13-1-0438 and W911NF-19-2-0049).

**Institutional Review Board Statement:** Not applicable.

**Informed Consent Statement:** Not applicable.

**Data Availability Statement:** All data included in this study are available from the corresponding author on reasonable request.

**Acknowledgments:** Yasong Li would like to thank Ruixuan Li and Changwei Li for their help with this article.

**Conflicts of Interest:** The authors declare no conflict of interest.

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
