# Peer review of "Microstructures and Properties of the Low-Density Al15Zr40Ti28Nb12M(Cr, Mo, Si)5 High-Entropy Alloys"

_metals, doi:10.3390/met12030496_

Round 1

Reviewer 1 Report

This manuscript investigates the effect of Cr, Mo and Si on the microstructure and mechanical properties of low density Al15Zr40Ti28Nb12M(Cr, Mo, Si)5 high-entropy alloys. The results are interesting and worth publication in Metals. However, there are several issues which need to be explained:

  • The scanning rate of 10 °/min is very high for an accurate XRD phase analysis.
  • How authors have prepared samples for EBSD and SEM studies are missing.
  • Why the intensity of three main BCC planes in XRD spectra changes dramatically for different alloys? There are no explanations about this in the manuscript.
  • For Mo5 sample, there is a diffraction peak around 41°, what is it? Generally, XRD test and analysis are performed superficial, and should be improved.
  • I recommend to discuss SEM and EDS results in same section of the manuscript.
  • The SEM image in Fig. 4; is it BSE or secondary electron image? Please indicate it.
  • Figure 6(e), the fracture surface of Al15Zr40Ti28Nb12Mo5, does not show the typical fracture surface of ductile materials with many dimples. The authors may provide a better image.

The English need to be improved. There are many grammatical and typo errors in the manuscript, for example:

“each sample is melting for at least 5 times…” should be “…was melted….”

“The microstructure of the back-scattered electron (BSE) photographs the…”

“…. and the with the Si addition…”

“ri is the ith element” should be “…is the radius…”

“…and ci and cj are the compositions of the ith and jth elements….” ci/cj is atomic fraction/percentage not composition.

Author Response

Response to Reviewer 1 Comments

Point 1: The scanning rate of 10 °/min is very high for an accurate XRD phase analysis.

Response 1: Thank you for your recomments. We replenished the XRD test with a lower scanning rate of 5 °/min. The results were shown in Figure 1(Figure 3 in the article).

Figure 1 The XRD partten of the Al15Zr40Ti28Nb12M5 alloys with scanning rate of 5 °/min.

Point 2: How authors have prepared samples for EBSD and SEM studies are missing

Response 2: Thank you for your recomments. We add some details in the section 2. Materials and Methods, with the prepartion of the samples for the EBSD and SEM.

Point 3: Why the intensity of three main BCC planes in XRD spectra changes dramatically for different alloys? There are no explanations about this in the manuscript.

Response 3: Thank you for your recomments. There may be some texture in the samples this results also happened in Ref. [1], shown in Figure 2(a). We retee the test with the same sample, as the result is as shown in Figure 2(b), and the resluts with another test samples were shown in Figure 1(Figure 3 in the article), the three main BCC planes in XRD spectra for different alloys is simliar.

Figure 2. Schematic diagram of composition-design philosophy. The internal XRD shows a single BCC lattice structure of as-cast (Zr0.5Ti0.35Nb0.15)100-xAlx alloys. (b) the XRD result with retee test.

Point 4: For Mo5 sample, there is a diffraction peak around 41°, what is it? Generally, XRD test and analysis are performed superficial, and should be improved.

Response 4: Thank you for your recomments. There may be some mistakes happend in the test, we retee the same sample by XRD with lower rate the peak at 41° was disapeared, shown in Figures 2(b). And the resluts with another Mo5 samples was test and the result were shown in Figure 1.

Point 5: I recommend to discuss SEM and EDS results in same section of the manuscript.

Response 5: Thank you for your recomments, we had made a discussion of the SEM and EDS results in same section.

Point 6: The SEM image in Fig. 4; is it BSE or secondary electron image? Please indicate it.

Response 6: Thank you. The SEM image in Fig. 4; is secondary electron image.

Point 7: Figure 6(e), the fracture surface of Al15Zr40Ti28Nb12Mo5, does not show the typical fracture surface of ductile materials with many dimples. The authors may provide a better image.

Response 7: Thank you, we had provide a better image.

Point 8: The English need to be improved. There are many grammatical and typo errors in the manuscript, for example:

“each sample is melting for at least 5 times…” should be “…was melted….”

“The microstructure of the back-scattered electron (BSE) photographs the…”

“…. and the with the Si addition…”

“ri is the ith element” should be “…is the radius…”

“…and ci and cj are the compositions of the ith and jth elements….” ci/cj is atomic fraction/percentage not composition.

Response 8: Thank you. Typo-errors and grammatical mistakes have been corrected in the manuscript.

Thank you very much for your time and consideration.

References:

[1] X. Yan, P.K. Liaw, Y. Zhang, Ultrastrong and ductile BCC high-entropy alloys with low-density via dislocation regulation and nanoprecipitates, Journal of Materials Science & Technology, 110 (2022) 109-116.

Reviewer 2 Report

The authors studied the microstructure and properties of 3 HEAs with Cr, Mo and Si on the same base of AlZrTiNb. I consider that the manuscript can be published only after some minor revisions as follows:

  • it is necessary to mention the preparation of the samples from the ingots for microscopy analysis.
  • title of the paragraph 3.2 must be corrected.

Author Response

Response to Reviewer 2 Comments

Point 1: The authors studied the microstructure and properties of 3 HEAs with Cr, Mo and Si on the same base of AlZrTiNb. I consider that the manuscript can be published only after some minor revisions as follows:

It is necessary to mention the preparation of the samples from the ingots for microscopy analysis.

Response 1: We had supplemented some details in the section 2. Materials and Methods, with the prepartion of the samples for microscopy analysis.

Point 2: Title of the paragraph 3.2 must be corrected.

Response 2: We had revised the title of the paragraph 3.2 Properities and compression fracture mechanism

Thank you very much for your time and consideration.

Reviewer 3 Report

- The writing of English should be improved, I suggest the authors get editing help from someone with full professional proficiency in English. Meanwhile, there are some spelling mistakes that should be carefully checked. 
For example; Al15Zr40Ti28Nb12SiMo5, and the with the Si addition, 45. (45) Yang, X. , Properities, The compressive stress-engineering strain, Al15Zr40Ti28Nb12Si5 HEA the alloy
- Abbreviations should be used to name the samples for ease of use in the text. Cr5, Si5, Mo5 are recommended.
- The subtitles of Figures 6 and 1 should be rewritten and abbreviated.
- More complete details of sample preparation, testing, and characterization should be provided
- State of the art to be improved further. The authors use mostly self-citation, it should be reduced.
- “The EBSD images and the grain size” are Should be given for “ Al15Zr40Ti28Nb12Si5 HEA”.
-Page 5; The value sequence line should be changed from "the compressive ductilities of these alloys are ~ 9%, ~ 7% and more than ~ 50%" to "~ 9%, ~ 50% and more than ~ 7%".
- The actual density of the samples must be determined.
- Silicide precipitates should be characterized by different methods. Doing this, reviewing the following ref could be helpful:
Materials Science and Technology, 30(4), pp.424-433.
- Preparation and characterization of Al20 (Zr50Ti35Nb15) 80 alloy is beyond the scope of this article and should be removed from the article and not included in the main results.
- Literature review is not sufficient and could be expanded. The following papers could be helpful:
Scripta Materialia 210, 114473
Journal of Alloys and Compounds 897, 163217

Author Response

Response to Reviewer 3 Comments

Point 1: The writing of English should be improved, I suggest the authors get editing help from someone with full professional proficiency in English. Meanwhile, there are some spelling mistakes that should be carefully checked.

For example; Al15Zr40Ti28Nb12SiMo5, and the with the Si addition, 45. (45) Yang, X. , Properities, The compressive stress-engineering strain, Al15Zr40Ti28Nb12Si5 HEA the alloy

Response 1: Thank you for your recomments. Typo-errors and grammatical mistakes have been corrected in the manuscript.

Point 2: Abbreviations should be used to name the samples for ease of use in the text. Cr5, Si5, Mo5 are recommended.

Response 2: Thank you for your recomments. We had used to name the samples for ease with Cr5, Si5, Mo5 in the text

Point 3: The subtitles of Figures 6 and 1 should be rewritten and abbreviated.

Response 3: Thank you for your recomments. We had rewritten and abbreviated the subtitles of Figures 6 and 1.

Point 4: More complete details of sample preparation, testing, and characterization should be provided.

Response 4: Thank you for your recomments. We had add some detailes for the sample preparation, testing, and characterization.

Point 5: State of the art to be improved further. The authors use mostly self-citation, it should be reduced.

Response 5: Thank you for your recomments, we had made a exchange to the self-citation, and add some more citations with others.

Point 6: “The EBSD images and the grain size” are Should be given for “ Al15Zr40Ti28Nb12Si5 HEA”.

Response 6: Thank you for your recomments. As there are a large number of silicides apeared in the microstructure of the Al15Zr40Ti28Nb12Si5 HEA, it is difficult for the preparetion of the samples for EBSD, we had try some methods to prepare the EBSD samples, unfortunately, failed.

Point 7: Page 5; The value sequence line should be changed from "the compressive ductilities of these alloys are ~ 9%, ~ 7% and more than ~ 50%" to "~ 9%, ~ 50% and more than ~ 7%".

Response 7: Thank you for your recomments. We had revised this mistake with "values of these alloys are ~ 9%, over ~50% and ~ 7%, respectively.

Point 8: The actual density of the samples must be determined.

Response 8: Thank you. We had made a test on the actual densities of the samples and add in the text.

Point 9: Silicide precipitates should be characterized by different methods. Doing this, reviewing the following ref could be helpful:

Materials Science and Technology, 30(4), pp.424-433.

Response 9: Thank you for your recomments. We read this paper carefully and it is very helpful to the characterization of silicides, and we had made a citation on the follwing ref.

Point 10: Preparation and characterization of Al20 (Zr50Ti35Nb15) 80 alloy is beyond the scope of this article and should be removed from the article and not included in the main results.

Response 10: Thank you for your recomments. Firstly, this study was based on the Al20(Zr50Ti35Nb15)80 alloy, and the preparation of large samples is the key to the application. And we found that the phase structure of the large ingot is same with the small mass one. Subsequently, the10kg-level ingots of Al15Zr40Ti28Nb12M5(Cr, Mo, Si) HEAs can also be prepared by this method, which provide a support for the flowing research works and industrialization application.

Point 11: Literature review is not sufficient and could be expanded. The following papers could be helpful:
Scripta Materialia 210, 114473

Journal of Alloys and Compounds 897, 163217

Response 11: Thank you for your recomments. We had made a citation on the follwing refs.

Thank you very much for your time and consideration.

Round 2

Reviewer 1 Report

The authors provided appropriate corrections to their manuscript. The revised manuscript can be published in Metals.

Reviewer 3 Report

The revised paper is acceptable for publication.